# A Novel Method for Recognizing Space Radiation Sources Based on Multi-Scale Residual Prototype Learning Network

**DOI:** 10.3390/s23104708

**Published:** 2023-05-12

**Authors:** Pengfei Liu, Lishu Guo, Hang Zhao, Peng Shang, Ziyue Chu, Xiaochun Lu

**Affiliations:** 1National Time Service Center, Chinese Academy of Sciences, Xi’an 710600, China; liupengfei@ntsc.ac.cn (P.L.); zhaohang@ntsc.ac.cn (H.Z.); shangpeng19@mails.ucas.ac.cn (P.S.); chuziyue@ntsc.ac.cn (Z.C.); luxc@ntsc.ac.cn (X.L.); 2University of Chinese Academy of Sciences, Beijing 100049, China; 3Key Laboratory of Precise Positioning and Timing Technology, Chinese Academy of Sciences, Xi’an 710600, China

**Keywords:** space radiation source, closed set recognition, open set recognition, prototype learning

## Abstract

As a basic task and key link of space situational awareness, space target recognition has become crucial in threat analysis, communication reconnaissance and electronic countermeasures. Using the fingerprint features carried by the electromagnetic signal to recognize is an effective method. Because traditional radiation source recognition technologies are difficult to obtain satisfactory expert features, automatic feature extraction methods based on deep learning have become popular. Although many deep learning schemes have been proposed, most of them are only used to solve the inter-class separable problem and ignore the intra-class compactness. In addition, the openness of the real space may invalidate the existing closed-set recognition methods. In order to solve the above problems, inspired by the application of prototype learning in image recognition, we propose a novel method for recognizing space radiation sources based on a multi-scale residual prototype learning network (MSRPLNet). The method can be used for both the closed- and open-set recognition of space radiation sources. Furthermore, we also design a joint decision algorithm for an open-set recognition task to identify unknown radiation sources. To verify the effectiveness and reliability of the proposed method, we built a set of satellite signal observation and receiving systems in a real external environment and collected eight Iridium signals. The experimental results show that the accuracy of our proposed method can reach 98.34% and 91.04% for the closed- and open-set recognition of eight Iridium targets, respectively. Compared to similar research works, our method has obvious advantages.

## 1. Introduction

Space situational awareness (SSA) refers to the ability to monitor, recognize and predict the identity, position and behavior of space targets, including debris and active satellites. References [1,2] pointed out that space situational information is crucial for ensuring the safety and sustainability of space operations. Therefore, space target recognition (STR) is a critical technology that can recognize potential threat targets, including debris or hostile satellites. Developing advanced STR capabilities is essential for maintaining the security and longevity of space missions.

The term “space targets” refers to a variety of satellites with various goals in this paper. These satellites can be used for communication, navigation, earth observation and scientific research purposes. They are an essential component of modern technology and play a significant role in our daily lives [2]. In the field of signal processing, these satellites can also be known as space radiation sources. However, non-cooperative, threatening or hostile satellites may also pose a significant risk to our security through some means, such as communication reconnaissance and intelligence collection. Therefore, it is important to continue monitoring and recognizing them. Additionally, understanding these space radiation sources can also help us improve our ability to quickly take countermeasures.

The signals emitted by space radiation sources to the ground usually exhibit significant nonlinear distortion because the high-power amplifier incorporated into the satellite transponder commonly operates in the saturated state or a state comparable to the saturated state to ensure high work performance [3]. In addition, due to the inherent differences in the manufacturing process of the internal components of the amplifier, the nonlinear distortion generated by the signal is different. It enables each space radiation source signal to carry unique fingerprint features different from other space radiation sources, which can be used for individual recognition and identity authentication. Therefore, it is of great significance to use the fingerprint information of electromagnetic signals to carry out the task of space radiation source recognition.

The individual recognition of radiation sources is essentially a pattern recognition problem [4]. In the early stage, the methods based on experience-driven strategy [5,6,7,8,9,10,11] take the leading position. They usually use traditional signal processing and expert feature-matching algorithms instead of recognition algorithms, which is difficult to achieve the expected accuracy in complex electromagnetic environments [12]. Recently, data-driven strategy based on massive data and deep learning has become popular. A lot of studies [13,14,15,16,17,18,19,20,21] have confirmed that deep learning has a strong feature extraction ability, which makes it superior to traditional methods in many recognition tasks. However, these works are mainly based on improved or new proposed network models. When Softmax Loss is used to train these models, they can only learn the inter-class separable features [22]. It may force us to increase the complexity of data preprocessing and network models in exchange for improving the recognition effect. Therefore, further enhancing the intra-class compactness of features may be a better choice.

In addition, the more realistic scenes we face are often open-set or open-world recognition [23], such as face recognition and automatic driving. Because new satellites will be launched continuously (at least at present), the space radiation source recognition task is more inclined to open set recognition. When judging the unknown radiation sources, the closed-set recognition methods are easy to fail. Fortunately, the problem of open set recognition has been paid attention to by researchers in many fields and given some works in recent years [24,25,26,27,28,29,30,31]. In addition, Refs. [32,33] describe the probability model theory and the latest research progress of open set recognition, respectively, which provide a broader idea to solve the problem. However, the research about open set recognition of radiation sources is in a nascent stage. To the best of our knowledge, the probability threshold methods [12,24,27] and generative adversarial network (GAN) [28,29] are mainly used to solve the problem. Because the probability threshold is difficult to select and the training process of GAN is complex, we urgently need new methods to meet the actual needs.

The center loss in face recognition is introduced into automatic modulation classification to learn discriminative features that are easier to classify [22]. Inspired by it, we hope to improve the recognition ability of the model by optimizing the new loss rather than designing complex networks. Due to the great success of prototype learning in image recognition [34,35], we propose a new space radiation source recognition method based on prototype learning in this paper. The method can be used to complete the closed- and open-set recognition of space radiation sources. Specifically, the main works and contributions of this paper are

(1)In order to improve the recognition accuracy of space radiation sources, we introduce the prototype learning strategy in image recognition into radiation source recognition and propose a new space radiation source recognition method;(2)Whether it is closed- or open-set recognition, learning separable and discriminative features is an important way to improve the network recognition effect. We design a multi-scale residual prototype learning network to recognize space radiation sources. Convolution kernels of different scales are used to learn the inter-class separable features. On the one hand, residual network structure can alleviate the problems of gradient disappearance and model degradation. On the other hand, it can fuse features of different scales. By optimizing the joint loss based on prototype learning, it can not only enhance the inter-class separability of features but also further enhance the intra-class compactness of features;(3)In order to solve the problem of open-set recognition of space radiation sources, we also designed a joint decision algorithm based on MSRPLNet. The algorithm uses two distance rules to make joint decisions on sample labels. Compared with the existing open set recognition methods, its advantages are that it can automatically set thresholds based on data distribution and has strong generalization ability;(4)We built a set of satellite signal observation and reception systems with a 13 m large-aperture antenna and collected eight Iridium satellite time and location (STL) signals. Experimental results confirm the effectiveness and reliability of the proposed method for closed-set recognition and open-set recognition. As far as the authors know, this study is the first to use real space signals received in the outdoor environment to carry out radiation source recognition work.

The rest of the paper is organized as follows. Section 2 introduces some research works. Section 3 describes the problem faced by space radiation source recognition. Section 4 introduces the proposed research method. Experimental results are given in Section 5. Section 6 is the conclusion.

## 2. Related Works

In this section, we first introduce the works of individual recognition of radiation sources based on experience-driven and data-driven techniques. Then we present the current research progress about open set recognition of radiation sources. Finally, we introduce the related works of prototype learning.

### 2.1. Recognizing Radiation Sources Using Experience-Driven and Data-Driven Techniques

Traditional radiation source individual recognition methods are mostly based on experience-driven. It first relies on the manual extraction of intuitive and reliable expert features from samples and then matches these features with all templates in the feature library. The most similar template attribute is the recognition result. Expert features are mainly divided into modulation domain features and waveform domain features [13]. Modulation domain features mainly include I/Q amplitude and phase imbalance, carrier frequency offset and modulation offset of received signal caused by small-scale hardware-level defects [5]. Common waveform domain features include compressed bispectrum [6], fractal box dimension [7], multi-dimension approximate entropy [8], Hilbert–Huang transform [9], empirical mode decomposition [10] and variational mode decomposition [11]. However, on the one hand, the experience-driven methods depend on the prior knowledge of the signal and the existing signal processing tools. On the other hand, the extracted expert features are easily affected by noise, and the recognition accuracy fluctuates greatly [20].

With the development of deep learning theory and parallel fast computing support provided by GPU in recent years, the recognition methods based on data-driven have become a research hotspot in this field. A lot of research works are devoted to using deep learning technology to automatically extract individual features of radiation sources. A deep, complex residual neural network was proposed to capture the fingerprint features of I/Q baseband signals of WiFi network card devices [13]. In [14], a convolutional neural network (CNN) was used to learn RF impairments contained in the transmitter signals and compared with traditional classifiers such as support vector machine and logistic regression. In [15], the authors ingeniously constructed a three-dimensional convolutional neural network by using the short-term spatio-temporal properties of the raw I/Q signal rather than the idea of processing time series. In the experiments of recognizing identical and heterogeneous transmitters, the accuracy is about 99%. However, the increase in convolution dimension makes the construction of the data set and network training process more complex. Ref. [16] developed a convolutional neural network framework using the time-domain complex baseband error signals. The framework achieves 92.29% recognition accuracy on seven commercial ZigBee devices. The application of machine learning in RF transmitter recognition was studied in [17]. The authors evaluated the recognition effect of support vector machines, conventional deep neural nets, convolutional neural nets and deep neural nets with multi-stage training on 12 transmitters. The latter has significant advantages in the recognition of large transmitter populations. In [18], the author collected the I/Q signals emitted by five individual radiation sources and adopted multiple different neural networks to obtain a satisfactory classification effect. Ref. [19] used the features extracted by the residual network and complex-valued residual network as the real and imaginary parts of the classifier, respectively. The method works well when there are fewer labeled samples. However, using two networks to extract features is time-consuming. In addition, in order to further improve the recognition accuracy of communication radiation sources, part of the research works [6], Refs. [20,21] combined the manually extracted expert features with deep learning to obtain fusion features. However, the essence of the above works is still only to extract inter-class separable features. They all ignore the intra-class compactness of features.

### 2.2. Open-Set Recognition of Radiation Sources

Whether based on experience-driven methods or data-driven methods, the works in Section 2.1 are discussed under the premise of closed set recognition. Different from other fields, such as the open-set recognition of images, there are few studies on the open-set recognition of radiation sources. At present, methods based on probability threshold and GAN have been proposed. Refs. [24,27] used the Softmax threshold and OpenMax threshold to recognize unknown radiation sources, respectively. Besides, two outlier detection methods and an autoencoder are introduced in [27]. However, outlier detection is difficult to separate known classes. Ref. [12] proposed a combined siamese neural network. The network is equivalent to establishing a binary anomaly detection classifier for each known radiation source. However, the detection threshold of each classifier only depends on experience. In [28], the authors designed a radio frequency adversarial learning (RFAL) framework based on GAN to recognize “trusted” transmitters and “adversarial” transmitters. The generative model generates false signals that are very similar to real signals by inferring parameter space and copying time-invariant features. The discriminator model recognizes the “trusted” transmitters by judging the real signals and the false signals. For identifying “adversarial” transmitters, RFAL can achieve about 99.9% accuracy. Ref. [29] pointed out that for wireless signal data sets, directly using the existing open-set recognition algorithms cannot obtain satisfactory results. The authors proposed a multi-task counterfactual GAN framework to capture the modulation features and fingerprint features of wireless communication signals. It can enhance the robustness of the model and the adaptability of open set recognition. Although the frameworks based on the GAN can achieve good results, it is difficult to apply to practical scenarios because of complex architecture and long training time.

### 2.3. Prototype Learning

Ref. [34] successfully applied prototype learning and deep learning to image recognition for the first time. In this paper, Yang et al. proposed a new learning framework called convolutional prototype learning (CPL) and introduced several loss functions based on prototype learning. CPL aims to solve the closure of the Softmax layer in traditional CNN models. The authors also showed that the loss functions based on prototype learning could be used to learn discriminative features. Later, Yang supplemented the relevant concepts of CPL and loss functions based on prototype learning from other perspectives [35]. He pointed out that prototype learning has obvious advantages in closed-set recognition, open-set recognition and incremental learning. The work in [25] is most similar to our research content. The author proposed the surrounding prototype loss (SPL) for radar high-resolution range profile (HRRP) open set recognition. SPL can further learn inter-class separable features. However, SPL introduces new hyperparameters, and the thresholds of open-set recognition are all empirical values.

## 3. Problem Description

The downlink signals broadcast by satellites to the ground are usually affected by the complex space channel environment. When the ground antenna is aimed at the satellite, the signal received by the antenna usually contains various noises and interference in addition to useful information. Discrete IF signal x(n) obtained by signal collector system can be modeled as
(1)$$x(n)=[s(n)*h(n)+w(n)]\cdot e^{j2\pi f_{m}n},n=1,2,\ldots ,N$$
where s(n) and h(n) represent useful information and channel impulse response, respectively. w(n) is defined as additional noise and interference. ∗ is the convolution operator. $f_{m}$ is the mixed frequency produced by the crystal oscillator of the mixer, and $j=\sqrt{-1}$. N is the number of sampling points of the signal.

The actual received satellite signal x(n) is real signal data. To make full use of the amplitude and phase information contained in the signal, we perform a Hilbert transform on x(n) to obtain its complex form hilbert[x(n)] and extract the I/Q signals, which are recorded as
(2)I(n)=Real{hilbert[x(n)]},n=1,2,…,N
(3)Q(n)=Imag{hilbert[x(n)]},n=1,2,…,N
where I(n) and Q(n) represent the operators of the real and imaginary parts of hilbert[x(n)], respectively.

K training samples $\left\{(x_{i},y_{i})|i=1,2,\ldots K\right\}$ with known labels can be constructed by I(n) and Q(n). $x_{i}$ and $y_{i}$ are the i data sample and its corresponding category label, respectively. Each $x_{i}$ can be recorded as
(4)$$x_{i}=\left(\begin{array}{cccc} I(\frac{N(i-1)}{K}+1), & I(\frac{N(i-1)}{K}+2), & \cdots , & I(\frac{Ni}{K})\\ Q(\frac{N(i-1)}{K}+1), & Q(\frac{N(i-1)}{K}+2), & \cdots , & Q(\frac{Ni}{K}) \end{array}\right)$$

As shown in Figure 1a, it is assumed that the original database of space radiation sources contains K known label samples, which are denoted as $\mathrm{TR}=\{(x_{i},y_{i})|i=1,2,\ldots K\}$. These samples come from M known radiation sources, and $y_{i}\in [1,M]$. As shown in Figure 1b, S new samples $\mathrm{TE}=\{(x_{j},y_{j})|j=1,2,\ldots S\}$ are from the above M known and other P unknown radiation sources, and $y_{j}\in [1,M+P]$. We hope that training TR can not only allow the recognition of unknown radiation source samples but also correctly classify radiation source samples.

Once we train a perfect neural network model f(θ) using TR, the model has a powerful feature extraction ability. When using TE for testing, the features z extracted by f(θ) can be represented as
(5)$$z=f(x_{j};\theta )$$
where $x_{j}$ is a test sample in TE, θ is the trained network parameters.

Closed set recognition gives the closed classification boundary in Figure 1c. It can correctly classify the categories agreed upon in advance but cannot expand new categories. This is because the back end of the model f(θ) uses a closed Softmax layer [33]. The Softmax layer is often connected to the last fully connected layer (FC) of the model f(θ). The number of neurons in the last FC is the number of known radiation sources. Therefore, the Softmax layer can only convert output features z of the last FC into the probability that the test sample $x_{j}$ belongs to each known radiation source, which makes it exhibit inherent closeness and unable to adapt to the open set world. The probability conversion function of the Softmax layer can be expressed as
(6)$$p(y=j)=\frac{e^{z_{j}}}{\sum_{m=1}^{M}e^{z_{m}}},j\in [1,M]$$
(7)$$\sum_{j=1}^{M}p(y=j)=1$$
where p(y=j) is the probability that the sample is classified as known class j. M is the number of known classes in the original database. $z_{j}$ is the j feature of the output z of the last FC. Because p needs to follow the probability distribution condition (7), the Softmax layer can easily classify unknown samples as one of the known radiation sources. In addition, the Softmax layer in the process of network training will gradually incline to the category with the largest probability distribution, which leads to the Softmax layer giving false high confidence more confidence.

Open set recognition gives the ideal open classification boundary in Figure 1d. It can not only recognize unknown radiation source samples but also correctly classify known radiation source samples. In a mathematical sense, open-set recognition requires us to find the measurable identification function f and minimize the joint risk [32]:(8)$$\underset{f}{\arg\min}C_{O}(f)+\lambda C_{\varepsilon }(f(V))$$
where joint risk consists of open space risk $C_{O}(•)$ and experiential risk $C_{\varepsilon }(•)$. $C_{O}(•)$ refers to the risk of labeling samples in an open space (an unknown space far from known radiation sources) as known radiation sources. $C_{\varepsilon }(•)$ represents the risk of confusion between known radiation sources. We need to minimize $C_{O}(•)$ and $C_{\varepsilon }(•)$ as much as possible to meet the requirement of open set recognition. V represents the training data and only contains known radiation source samples. λ is a user-defined constant used to balance the two risks. It requires recognizing unknown radiation source samples on the basis of correctly recognizing known radiation source samples.

## 4. Proposed Method

In this section, we first introduce the framework of the proposed space radiation source recognition method. Compared with existing radiation source recognition works, our method can be used for both closed- and open-set recognition tasks. Then we give a detailed description of the proposed MSRPLNet. MSRPLNet improves the recognition performance of radiation sources by enhancing the inter-class separation and intra-class compactness of sample features, while the novel models designed in most of the other works only improve the inter-class separation of features. Finally, we introduce the proposed joint decision algorithm for unknown radiation source recognition. The algorithm solves the problem of manually setting empirical thresholds and provides better generalization ability compared to traditional methods.

### 4.1. Space Radiation Source Recognition Method Based on MSRPLNet

By introducing the prototype learning strategy, we propose a new space radiation source recognition method. Unlike the works in [12,28,29], our method pays more attention to learning separable features and discriminative features. The method can be used for the closed- and open-set recognition of space radiation sources. Figure 2 shows the framework of our proposed method, including data preprocessing, network training, feature extraction and recognition modules. In the data preprocessing module, raw I/Q signals are first sliced and normalized to construct the training set, validation set and testing set. The training set and validation set need to be labeled to learn the discriminative model. In the network training and feature extraction module, MSRPLNet is used to extract features with intra-class compactness and inter-class separability. When MSRPLNet training is completed, we can obtain the optimal model parameters, prototype representation and feature representation of the validation set. In the recognition module, when MSRPLNet extracts features from the testing set, the prototype representation and prototype matching can be used to recognize the closed set of space radiation sources. In addition, when the testing set contains unknown radiation source samples, the joint decision algorithm based on MSRPLNet can not only classify known radiation source samples but also detect and reject unknown radiation source samples.

### 4.2. Multi-Scale Residual Prototype Learning Network (MSRPLNet)

#### 4.2.1. Model Structure of MSRPLNet

As shown in Figure 3, a multi-scale residual prototype learning network MSRPLNet is proposed in this section. It is composed of an input layer, a convolutional layer, a batch normalization layer, an average pooling layer, an ReLU activation layer, a dropout layer, a multi-scale residual block, a flatten layer and a fully connected layer. The input of MSRPLNet is raw IF I/Q signal slices after standard normalization. We first use a convolutional layer with a 1×9 kernel to extract features from raw data. A 1×9 convolutional kernel has a larger receptive field, which can better capture spatial information contained in the data and extract global features. The batch normalization layer is used to standardize the features of each channel to promote the stable distribution of features and accelerate the learning speed of the network. Considering that the length after using the flatten layer to pull the extracted features into one dimension is too large, we take the average pooling operation on the feature maps in the early stage to reduce the amount of calculation and the required memory. The pooling size of 2×2 means that the features in each channel will be merged into one channel, and the number will be quartered. The ReLU activation layer is used to process the output of the upper neurons nonlinearly and transmit it to the lower neurons. The dropout layer prevents the model from overfitting by discarding some neurons. Then we input the extracted shallow features into multiple continuous multi-scale residual blocks. As shown in Figure 4, the construction of multi-scale residual blocks is the same, and their design inspiration comes from the residual unit in the residual neural network. In multi-scale residual blocks, input x will perform four convolutional block operations. Each convolutional block contains a convolutional layer, a Batch normalization layer, an ReLU activation layer and a Dropout layer. The difference between each convolutional block operation is the use of different convolutional kernels. The first route feature is extracted by two 1×7 kernels, the second route feature is extracted by two 1×5 kernels, the third route feature is extracted by two 1×3 kernels and the fourth route feature is the original feature, which can be considered to be extracted by two 1×1 kernels. Convolutional kernels of different sizes have different receptive fields, and they can capture more different features. Four route features are transformed into one route feature through the Add layer. On the one hand, Add layer inherits the advantages of the residual unit and can be used to alleviate the problems of gradient disappearance and model degradation. On the other hand, Add layer fuses four route features to obtain more separable features. Features Y extracted from a multi-scale residual block can be recorded as
(9)$$Y=\mathrm{ReLU}(Y_{1}⊕Y_{2}⊕Y_{3}⊕Y_{4})$$
(10)$$Y_{1}=F_{\beta =(1,7)}(F_{\beta =(1,7)}(x))$$
(11)$$Y_{2}=F_{\beta =(1,5)}(F_{\beta =(1,5)}(x))$$
(12)$$Y_{3}=F_{\beta =(1,3)}(F_{\beta =(1,3)}(x))$$
(13)$$Y_{4}=x$$
where $F_{\beta }(x)$ performs the convolutional block operation, which consists of a convolutional layer with a 1×β kernel, a batch normalization layer, an ReLU activation layer and a dropout layer with a discard rate of 0.5. x is the input of the multi-scale residual block. ⊕ represents the add operator. $Y_{1},Y_{2},Y_{3}$ and $Y_{4}$ represent the first route feature, the second route feature, the third route feature and the fourth route feature, respectively.

After passing through several successive multi-scale residual blocks, MSRPLNet can extract deep features of signals from different space radiation sources. The number of multi-scale residual blocks used depends on the learning ability of MSRPLNet. Finally, we use a flatten layer and three fully connected layers with different numbers of neurons to integrate the extracted abstract features and obtain the output features.

#### 4.2.2. Training Method of MSRPLNet

MSRPLNet has multi-scale residual blocks that can be used to fuse different levels of separable features, which improves the inter-class recognition effect. However, when the inter-class distance is less than the intra-class distance, the intra-class diversity will reduce the recognition effect. As shown in Figure 5, MSRPLNet introduces distance cross-entropy loss (DCE Loss) and prototype loss (PL Loss) for model training to further enhance the inter-class separation and intra-class compactness of features. MSRPLNet will maintain and learn several prototypes for each class. We denote the prototype as $m_{ij}$, where i∈{1,2,…,M} represents the category index, M is the number of categories, j={1,2,…,K} represents the index of the prototypes in each class and K is the number of prototypes in each class. Prototypes can be seen as abstract representations of features and learned along with features. Joint Loss based on prototype learning is weighted by DCE Loss and PL Loss. During the training process of MSRPLNet, the Adam optimizer is used to optimize Joint Loss to update network parameters and prototypes. In addition, when the current recognition accuracy of the validation set is lower than before, the learning rate l is multiplied by a decay factor r.

The same as Softmax Loss, DCE Loss also uses the Softmax layer to calculate the probability that the sample x belongs to the prototype $m_{ij}$. From this, we can get
(14)$$p(x\in m_{ij}|x,f)=\mathrm{Softmax}[-d(f(x),m_{ij})/\gamma ]$$
where $d(f(x),m_{ij})={\left\|f(x)-m_{ij}\right\|}_{2}^{2}$ represents the Euclidean distance between the sample feature f(x) and the prototype $m_{ij}$. γ is a hyperparameter used to control distance hardness.

Thus, the DCE Loss of sample x belonging to class y can be expressed as
(15)$$l_{DCE}((x,y);f;M)=-\log[\sum_{j=1}^{K}p(x\in m_{yj}|x,f)]$$

Considering that directly optimizing DCE loss may lead to model overfitting, we introduce PL loss as a regularization term to improve the generalization ability of the model. PL Loss is defined as
(16)$$l_{PL}((x,y);f;M)={\left\|f(x)-m_{yj}\right\|}_{2}^{2}$$
where $m_{yj}$ represents the nearest prototype of the class y corresponding to f(x).

By weighting DCE Loss and PL Loss, we further obtain Joint Loss:(17)$$l_{\mathrm{Joint}}((x,y);f;m)=l_{\mathrm{DCE}}((x,y);f;m)+\lambda \cdot l_{\mathrm{PL}}((x,y);f;m)$$
where λ is a hyperparameter used to balance DCE loss and PL loss.

According to the formulation in [34], Joint Loss is derivable for both prototypes and network parameters. It means that we can obtain the optimal network parameters and prototypes by optimizing Joint Loss. DCE Loss inherits the advantages of Softmax Loss, which can be used to learn the inter-class separable features. PL Loss enhances intra-class compactness by penalizing the Euclidean distance between features and their real prototypes. It plays a decisive role in improving the effectiveness of radiation source recognition.

#### 4.2.3. Decision Method of MSRPLNet

Traditional neural networks directly use the Softmax layer to calculate the classification probability after extracting features, while prototype learning is different, and it completely abandons the Softmax layer. Figure 6 shows the process of classifying samples using the prototype matching method during the recognition stage. When MSRPLNet training is completed, we first need to calculate the Euclidean distances $D=\{d_{ij}|i=1,2,\ldots M;j=1,2,\ldots K\}$ between the features extracted by MSRPLNet to the testing sample x and each prototype $m_{ij}$. Then find the smallest distance $d_{\min}$ in the D. The prototype category i corresponding to $d_{\min}$ is the label of the testing sample x. Suppose that f(x;θ) is a well-designed feature extractor and θ represents the learned network parameter. The prototype matching process can be expressed as follows:(18)$$x\in class,\overset{M}{\underset{i=1}{\arg\max}}g_{i}(x)$$
where $g_{i}(x)$ is the discriminative functions of class i, and it represents the similarity between sample x and class i. $g_{i}(x)$ is denoted as
(19)$$g_{i}(x)=-\overset{K}{\underset{j=1}{\min}}{\left\|f(x;\theta )-m_{ij}\right\|}_{2}^{2}$$

### 4.3. Joint Decision Algorithm

In the prototype matching method, when the Euclidean distance between a sample feature and a type of prototype is the smallest, the class of the prototype is assigned to the sample. However, prototype matching cannot be used to recognize unknown radiation sources. It will still recognize the unknown sample as the class corresponding to the most recent prototype. Therefore, we propose a joint decision algorithm based on prototype distance and center distance to recognize unknown space radiation sources in Figure 7 and Algorithm 1.
**Algorithm 1** Open set recognition of MSRPLNet.**Input:** Training set xtrain, Validation set xvalid, Testing set xtest, Initialized learning rate l=0.001. Initialized prototypes mij, hardness parameter γ, balance parameter λ, network parameter θ, maximum epochs and the number of “Early Stopping.” Initialized learning rate decay factor r and the number of iterations t=0. **Output:** The parameter θ, prototypes mij and testing set labels ytest**Stage 1:**
**While** the stopping criterion does not meet, **do**
    t=t+1     Select a min-batch m sample from xitrain and feed it into MSRPLNet     Compute the Joint Loss lJoint=lDCE+λ·lPL    Compute two gradients by error back-propagation and chain rule: ∂lJoint/∂θ and ∂lJoint/∂mij    Update θt+1 by θt+1=θt−l·(1−rt)·(∂lJointt/∂θt)     Update mijt+1 by mijt+1=mijt−l·(1−rt)·(∂lJointt/∂mijt)**End while**
**Return**
θ, mij, f(xvalid;θ) and f(xtest;θ)**Stage 2**: **For**
k
**=** 1: number(xcorrectvalid) Compute f(xkvalid;θ)−mij22   Compute ci=1n∑n=1Nf(xcorrectivalid;θ) and f(xkvalid;θ)−ci22**End**
Find prototype distance dmi and center distance dci**Return**
dmi, ci and dci**Stage 3:**
**For**
b
**=** 1: number(xtest)    Compute f(xbtest;θ)−mij22 and find the minimum prototype distance dmb   Compute f(xbtest;θ)−ci22 and find the minimum center distance dcb   Let δ1=dmi and δ2=dci**If**
dmb≤δ1 and dcb≤δ2: xbtest is the known target sample and ybtest=i**Else:**
xbtest is the unknown target sample and ybtest=unknown**End**
**End**
**End**
**Return**
$y^{test}$

As shown in Figure 7, we assume that each class has only one prototype. Let the feature of the sample x extracted by MSRPLNet is f(x). The prototype with the smallest Euclidean distance from f(x) is $m_{i}$. The center feature of class i corresponding to $m_{i}$ is $c_{i}$. $c_{i}$ is given by the average value of the sample features recognized correctly for each class in the validation set. Given two thresholds $\delta_{1}$ and $\delta_{2}$, when the following relationship is satisfied
(20)$$\left\{\begin{array}{c} {\left\|f(x)-m_{i}\right\|}_{2}^{2}\leq \delta_{1}\\ {\left\|f(x)-c_{i}\right\|}_{2}^{2}\leq \delta_{2} \end{array}\right.$$

We can think that the sample x belongs to class i. Otherwise, it is from unknown radiation sources. The most important problem is how to determine $\delta_{1}$ and $\delta_{2}$. We give the following method to set two thresholds automatically:(1)When the network verification accuracy reaches the expected level, it means that the network training is complete. For any sample x that is correctly recognized in the validation set, its corresponding feature f(x) and the most recent prototype $m_{i}$ are output and placed in class i;(2)After the classification is completed, the center feature $c_{i}$ of the class i is calculated;(3)Find the sample that is furthest from its prototype in each class of correctly recognized samples and calculate the corresponding prototype distance $d_{m_{i}}$;(4)Find the sample that is furthest from its center in each class of correctly recognized samples and calculate the corresponding center distance $d_{c_{i}}$;(5)Let $\delta_{1}$ = $d_{m_{i}}$, $\delta_{2}$ = $d_{c_{i}}$. We can automatically get $\delta_{1}$ and $\delta_{2}$ from the data.

In the testing stage, MSRPLNet is first used to extract feature f(z) of the testing sample z. Then the distance $d_{mb}$ from f(z) to its nearest prototype and the distance $d_{cb}$ from f(z) to its nearest center are calculated. If
(21)$$\left\{\begin{array}{c} d_{mb}\leq \delta_{1}\\ d_{cb}\leq \delta_{2} \end{array}\right.$$

The testing sample z is from known space radiation sources and is classified as class i. In any other case, it is determined that the testing sample z belongs to unknown classes.

The advantages of the joint decision algorithm of unknown space radiation sources based on prototype distance and center distance are (1) The algorithm can automatically set the thresholds. Thresholds are derived directly from data distribution without having to be set manually. (2) The features of unknown samples are likely to be close to a certain class of prototype in the feature space. It will cause the inter-class distance to be less than the intra-class distance and make a wrong decision. The introduction of center distance makes up for this shortcoming. (3) When the data distribution of the radiation source signal changes, the thresholds can be adjusted adaptively. It can improve the generalization ability of Algorithm 1.

## 5. Experiments and Results

In this section, we test the proposed space radiation source recognition method using eight Iridium signals acquired in the real environment. The experimental content is divided into two parts: closed-set recognition and open-set recognition. All experiments are implemented on a situational awareness workstation with an Intel Xeon W-2133 CPU and an NVIDIA Quadro P2000 GPU. The construction and testing of MSRPLNet are based on the Python and TensorFlow deep learning frameworks.

### 5.1. Data Collection

In order to truly verify the effectiveness and reliability of the proposed method, we build a set of satellite signal observation and receiving systems in the Xi’an Experimental Field of the National Time Service Center, Chinese Academy of Sciences. Figure 8 shows the main equipment used in the system, including a spectrum analyzer, an L-band 13 m large-aperture antenna, a signal collector system and a signal processing workstation. We first calculate the tracking time, azimuth and pitch angle of the antenna based on the TLE data of the Iridium. The above information is then imported into the antenna’s computer so that the antenna automatically tracks the position of the Iridium and receives the maximum gain signal. We use the spectrum analyzer to observe and search for Iridium signals in the L-band. When the signal frequency range is determined, the signal collector system is used to acquire the Iridium signal in this frequency range. The subsequent processing of the Iridium signal is completed on the workstation.

We collected eight Iridium STL signals for experimental validation. Table 1 gives some information about Iridium and its STL signal. The signal-to-noise ratio (SNR) range of the Iridium STL signals is about 10~14 dB by using the M2M4 algorithm. Because the wide-band signal collector system we use may collect multiple frequency signals, we perform parametric analysis and spectrum detection of all received Iridium signals before the experiment and separate the target signals. Figure 9 shows the spectrogram of an Iridium STL signal after smoothing filtering.

### 5.2. Data Set Construction

Ref. [36] and Iridium STL signal preamble verification work tell us that the preamble part of the signal is sufficient to distinguish different radiation sources and performs well. This is because the preamble usually contains important information that the receivers can quickly capture and track satellite signals. Figure 10 shows the time-domain waveform after demodulation of an Iridium STL signal. We can see that the preamble occupies the first 2.6 ms of the baseband signal. Therefore, we construct the data set using only the first 2.6 ms tone signal of all Iridium STL signals. The data used in the experiment are all IF I/Q signals, which are not down-converted and demodulated to baseband signals. It makes more practical sense. Because for most space radiation source signals, we have almost no prior knowledge and cannot demodulate them.

We use the signal collector system to collect 2 s data for each Iridium signal. The sampling frequency of the signal collector system is 250 MHz, and each Iridium STL signal contains 500 M sampling points. After data preprocessing, the number of samples of each Iridium STL signal is 2700, and the sample length is 4000. The total number of samples is 21,600. Each sample is standard normalized before being fed into the neural network. In all closed-set experiments, the data set is divided into a training set, a validation set and a testing set. Their split ratio is 0.64:0.16:0.2.

### 5.3. Closed Set Recognition

In this section, the proposed method is verified based on the data set constructed by eight Iridium STL signals. MSRPLNet has an initial learning rate of 0.001, a maximum epoch of 50 and a batch size of 64. If the current recognition accuracy of the validation set is lower than before, the learning rate is multiplied by a decay factor. In addition, the “Early Stopping” strategy is used in the network training process to prevent model overfitting. We first discuss the hyperparameters that affect MSRPLNet recognition performance and compare MSRPLNet with common models of different network structures and loss functions. We also analyze the degree of confusion in the recognition of eight Iridium targets. In the following experiments, each set of data in the boxplot and histogram is the result of multiple experiments.

In order to explore the influence of network structure hyperparameters on MSRPLNet recognition performance, Figure 11a,b show the recognition results of the number of convolution kernels and multi-scale residual blocks on the above eight Iridium targets, respectively. In Figure 11a, according to the average recognition accuracy in the boxplot, it can be seen that when N>12, the improvement in average recognition accuracy of eight Iridium targets is no longer significant. In Figure 11b, we can see that the number of the multi-scale residual block M has a significant effect on the recognition accuracy of eight Iridium targets. When M=1, MSRPLNet performs best and achieves an average recognition accuracy of 98.24%, which is 1.09% higher than when M=0. Therefore, multi-scale residual blocks can further extract separable features. When M>1, the model overfitting caused by network deepening reduces the recognition accuracy of MSRPLNet. Because MSRPLNet adopts a residual network structure, the downward trend is relatively slow and alleviates the overfitting problem to some extent. Due to the high SNR of the Iridium STL signals received by the high-gain antenna, we can also obtain a good recognition effect when using a small number of convolution kernels and multi-scale residual blocks.

In order to explore the influence of network training hyperparameters on MSRPLNet recognition performance, Figure 12a,b show the recognition results of the learning rate decay factor r and the hardness parameter γ and balance parameter λ of Joint Loss on the above eight Iridium targets, respectively. In Figure 12a, MSRPLNet is difficult to converge quickly when r=0 because the learning rate is too high. The average recognition accuracy of MSRPLNet only reaches 97.16%. When r=0.8, the recognition performance of MSRPLNet decreases because of the fast decay of the learning rate and the slower learning speed of the model. When r=0.4, MSRPLNet achieves the best recognition effect. In Figure 12b, we label the hardness parameter γ and the balance parameter λ as data pair (γ,λ). (γ,λ) can be used to control the distance between features and prototypes, and it determines the inter-class separation and intra-class compactness of features. When γ=0.5,λ≠0, the recognition accuracy of MSRPLNet decreases with the increase of λ. This is because the feature space is too compact, and the overfitting problem is due to the increase in the proportion of DL Loss. When γ<5,λ=0, the overall performance of MSRPLNet is significantly reduced. MSRPLNet achieves the best recognition accuracy of 98.34% when (γ,λ)=(5,0). Therefore, the selection of (γ,λ) largely determines the closed-set recognition performance of MSRPLNet.

In order to verify the influence of different models on the recognition performance of eight Iridium targets, Figure 13 compares the recognition performance of MSRPLNet with several representative models, including DNN, 2DCNN, RNN-LSTM, ResNet, CNN/VGG and CLDNN. Except for the different network structures, all models use the Joint Loss and prototype matching methods. In addition, the back end of all models employs three fully connected layers with 128, 16 and eight neurons, respectively, to extract output features. Table 2 shows the training parameters of MSRPLNet. In Figure 13, MSRPLNet undoubtedly achieves the best result. ResNet alleviates the model-overfitting problem due to its “shortcut connection” structure, which makes its average recognition probability reach 98.13%. Comparing MSRPLNet and ResNet, we can still see that multi-scale residual blocks can extract more separable features. Compared with DNN, 2DCNN, CLDNN, CNN/VGG and RNN-LSTM, the recognition accuracy of MSRPLNet is increased by 1.06%, 0.48%, 1.66%, 1.36% and 1.98%, respectively. It further demonstrates the superiority of the MSRPLNet. Table 3 compares the time consumption when different models are trained once. Due to the use of multi-scale residual blocks, MSRPLNet training takes a long time. CLDNN consists of multiple CNNs and LSTMs. Increased network depth means that training is more time-consuming. In contrast, DNN takes the least time because it only contains three fully connected layers.

In order to verify the influence of different losses on the recognition performance of eight Iridium targets, Figure 14 compares the recognition results using Softmax Loss and Joint Loss during training. Here we use three models: CLDNN, 2DCNN and MSRPLNet. When MSRPLNet is trained with Softmax Loss, we define the network name as MSRNet. It is clear that the three models achieve the best recognition results when using Joint Loss. For CLDNN, 2DCNN and MSRNet, the recognition accuracy of Joint Loss is 0.42%, 0.39% and 0.71% higher than Softmax Loss, respectively. It shows that Joint Loss can optimize and improve the performance of the model. In Figure 15, we use the PCA method to visualize the output features of the last fully connected layer and prototypes of the three models and convert them into a two-dimensional scatter map. As shown in the first row, all three models can learn inter-class separable features when Softmax Loss is used in the training process. However, although the inter-class separation is obvious, there still exists an overlap between different target features. When Joint Loss is used, the distance between the prototypes will gradually increase with the training process because the prototypes and the network parameters are jointly learned. It makes the inter-class separation more obvious and also means more compact within the class. For example, the features of target 5, target 6, and target 7 become more compact when 2DCNN is combined with Joint Loss. It is quite evident that MSRPLNet with Joint Loss works best. In addition, for different models and loss functions, some features overlap between target 4 and target 5, between target 6 and target 8 and between target 2 and target 7, which means that the above three sets of Iridium targets may be difficult to classify.

In order to further understand the difficulty of closed set recognition of eight Iridium targets, Figure 16a,b show the confusion matrix of MSRNet + Softmax Loss and MSRPLNet, respectively. When MSRNet and Softmax Loss are trained together, the recognition accuracy of all Iridium targets except target 4 reaches more than 97%. When MSRPLNet is used for training, the recognition accuracy of all Iridium targets except target 4 reaches more than 98%. The result of the confusion matrix shows that MSRPLNet has better recognition performance on eight Iridium targets. In addition, we can see that target 1 and target 3 have the best recognition effect, and their recognition accuracy can reach 100%. Target 4 has the worst recognition accuracy at only 95%. As shown in Figure 16a, target 5 is most likely to be confused with target 4, target 8 is most likely to be confused with target 6, and target 7 is most likely to be confused with target 2. This is consistent with our analysis above.

### 5.4. Open Set Recognition

In this section, open-set recognition experiments of eight Iridium targets are carried out based on the proposed joint decision algorithm of unknown space radiation sources and MSRPLNet. We take 20% of the samples of each known target and all samples of the unknown targets as the testing set. The remaining 80% of the samples of each known target are used as the training set and validation set with a ratio of 0.64:0.36. All unknown targets are added sequentially according to signal acquisition time. We first discuss the influence of (γ,λ) on the open set recognition performance. To further illustrate the superiority of the proposed method, we compare it with other open-set recognition works by ablation experiments. In addition, we also verify the ability of the proposed method through two individual evaluation experiments. As with most open-set recognition works, we introduce the concept of openness [23] to define open space
(22)$$openness=1-\sqrt{\frac{2\times N_{train}}{N_{train}+N_{test}}}$$
where $N_{train}$ is the number of known targets in the training set. $N_{test}$ is the number of targets to be recognized in the testing set. The larger $N_{train}$, the smaller openness.

In order to explore the influence of the hardness parameter γ and balance parameter λ on the open set recognition performance of the proposed method, Table 4 shows the average recognition accuracy of the testing set under different openness and (γ,λ). openness=52.86% means that there are only samples of a known target in the training set, and there are samples of this known target and seven other unknown targets in the testing set. The interpretation of the remaining openness can be based on this analogy. In Table 4, the proposed method works best for the open set recognition of eight Iridium targets when (γ,λ)=(5,0.001) or (γ,λ)=(5,0.01). When λ>0, because DL Loss forces the model to learn more discriminative features, the open set recognition effect is greatly improved compared with λ=0. When γ=0.5 and γ=5, we can see that the increase in γ also improves the open set recognition effect under different openness. Therefore, the selection of (γ,λ) also determines the open set recognition performance of MSRPLNet and the joint decision algorithm. Additionally, our proposed MSRPLNet and joint decision algorithm can achieve the recognition accuracy of 91.04% for open set recognition of eight Iridium targets.

In order to further understand the influence of (γ,λ) on the recognition performance of the joint decision algorithm, Figure 17a–c give the recognition accuracy of known Iridium target samples and unknown Iridium target samples in the testing set respectively when (γ,λ)=(0.5,0), (γ,λ)=(0.5,0.01) and (γ,λ)=(5,0.01). We can find that when λ=0, the proposed joint decision algorithm has a better recognition effect on known Iridium targets than on unknown Iridium targets. It shows that MSRPLNet only pays attention to the inter-class separation of known targets and ignores the intra-class compactness. Therefore, the joint decision algorithm makes the wrong decisions on many unknown Iridium target samples. When γ increases from 0.5 to 5 and λ increases from 0 to 0.01, the recognition accuracy of the proposed joint decision algorithm for unknown Iridium targets gradually increases. It shows that enhancing the inter-class separation and intra-class compactness of features can significantly improve the open set recognition performance of the joint decision algorithm. In addition, when openness=52.86%, the recognition accuracy of unknown Iridium targets is low. It reflects that too large openness will make the open set recognition more difficult.

Next, we compare the proposed method with some current open-set recognition works. It is worth noting that we do not contrast it with GAN because we hope that the model has a simple training process in practical application. The Softmax threshold [12] is the simplest baseline method. OpenMax [13] is the first formally proposed method to solve the problem of open set recognition. The closest to our work are the PL, GCPL and SPL proposed in [15]. The three methods are all based on prototype learning, and the difference is that the model adopts different losses. PL only uses DCE Loss, GCPL uses Joint Loss, and SPL adds a surrounding constraint term to Joint Loss. Above all, the five open-set recognition methods used for comparison are all based on Softmax probability. Therefore, an additional judgment threshold is required. Unfortunately, these five works give empirical thresholds appropriate to the data they process. It does not guarantee that these thresholds will still be available when applied to other data. Our approach abandons the Softmax layer and makes decisions based on distance. In addition, the distance thresholds in the proposed method are only related to the feature distribution of the signals. This allows them to be set automatically and adaptively.

Figure 18a,b compare the recognition performance of the above open-set recognition works on known Iridium target samples and unknown Iridium target samples in the testing set, respectively. The open-set recognition methods based on Softmax probability all use the threshold τ=0.98 recommended in [24]. It is clear that τ=0.98 is not suitable for our data in some openness. Softmax, OpenMax and PL work well for known Iridium target recognition, but it is not satisfactory for unknown Iridium target recognition. GCPL and SPL may fail when recognizing known Iridium targets. We attempt to determine a good threshold, but the output of the Softmax layer is too confident. In addition, we find that different network initialization parameters also affect the output probability of the Softmax layer. This adds difficulty to the selection of threshold τ. When openness<52.86%, our proposed joint decision algorithm is more efficient. When openness=52.86%, all methods fail to recognize unknown Iridium targets because of the maximum openness. Even so, our method still has better generalization ability than other methods.

In order to further illustrate the effectiveness of the proposed method for the open-set recognition of eight Iridium targets, we perform two individual evaluation experiments. The first experiment shows that MSRPLNet can obviously improve the open set recognition accuracy. The testing set contains the samples of four known Iridium targets (A~D corresponds to target 1~4) and four unknown Iridium targets (unknown targets correspond to 5~8) when openness=18.35%. The number of feature points for each known Iridium target is 540, and the number of feature points for each unknown Iridium target is 2700. As shown in Figure 19a,b, we used the PCA method to convert features and prototypes into a two-dimensional scatter map. In Figure 19a, it is clear that the four known Iridium target features are highly separated, while the unknown Iridium target features are almost overlapped with all known Iridium target features. MSRNet + Softmax Loss can only learn the inter-class separable features, which makes it more difficult to solve the problem of open set recognition in real scenarios. In Figure 19b, it is clear that MSRPLNet can learn the inter-class separable features and intra-class discriminative features at the same time. It benefits from the use of multiscale residual blocks and Joint Loss. In addition, we can also see the fact that MSRPLNet does not divide the entire feature space like MSRNet + Softmax Loss but instead projects features near prototypes. It provides a good idea to detect and reject unknown targets. However, because the signals used in the experiments are from multiple Iridium targets with high similarity, some unknown Iridium target features still overlap with the features of known Iridium targets 2 and 4.

The second experiment proves that the joint decision algorithm is superior to the single decision algorithm. We still consider the situation when openness=18.35%. Figure 20a,b show the confusion matrix when using the single decision algorithm and the joint decision algorithm, respectively. The difference between the two algorithms is that different distance rules are used. Figure 20a only considers the prototype distance, while Figure 20b uses the prototype distance and the center distance. We can find that no matter which distance rule is used, there is no confusion between the known Iridium target features. Because the unknown Iridium target features are likely to be close to a certain type of prototype, some unknown Iridium target samples may be misidentified as known Iridium targets. In addition, the use of distance rules can also cause the known Iridium target samples to be misidentified as unknown Iridium targets. When the prototype distance and center distance are jointly used, we can see that the recognition accuracy of both known and unknown Iridium targets will be improved. It reflects the effectiveness and reliability of the proposed joint decision algorithm in open set recognition of eight Iridium targets.

## 6. Conclusions

In this work, we propose a new prototype learning method for the closed-set recognition and open-set recognition of space radiation sources. The idea is to improve the recognition accuracy of radiation sources by enhancing the intra-class compactness and inter-class separability of the learning features. To this end, we designed a multi-scale residual prototype learning network, MSRPLNet. Aiming at the recognition problem of unknown radiation sources, we propose a joint decision algorithm based on prototype distance and center distance. The algorithm relies only on the feature distribution of the data without setting empirical thresholds. It can also improve the generalization ability of open set recognition. Based on multiple experimental results on eight Iridium STL signals, we confirm that the proposed method has obvious advantages for both closed-set recognition and open-set recognition. Under the condition of closed set recognition, MSRPLNet and prototype matching can achieve an accuracy of 98.34% for eight Iridium targets. Under the condition of open set recognition, MSRPLNet and the joint decision algorithm can achieve the highest unknown target recognition accuracy of 91.04%. However, there are still some potential deviations or limitations that may affect the effectiveness of this study. We list them as follows:(1)The extremely complex dynamic space environment seriously affects the quality of received signals. We need to improve the robustness of the proposed method to complex space environments;(2)The short duration of low earth orbit satellite signals results in a smaller number of samples. We need to develop new methods for recognizing space radiation sources under small sample signal conditions;(3)The difference in data distribution of the same target signal at different time periods may lead to model failure. We need to study signal fingerprint features that are relatively insensitive to time changes;(4)The use of different hardware devices (such as an antenna and signal collector, etc.) may affect the recognition performance of the model. We need to eliminate the impact of receiving devices on signal fingerprint features;(5)We need to consider how to update the model with new target data without retraining the model.

The above issues require us to think about and solve them in our future research work.

## Figures and Tables

**Figure 1 sensors-23-04708-f001:**
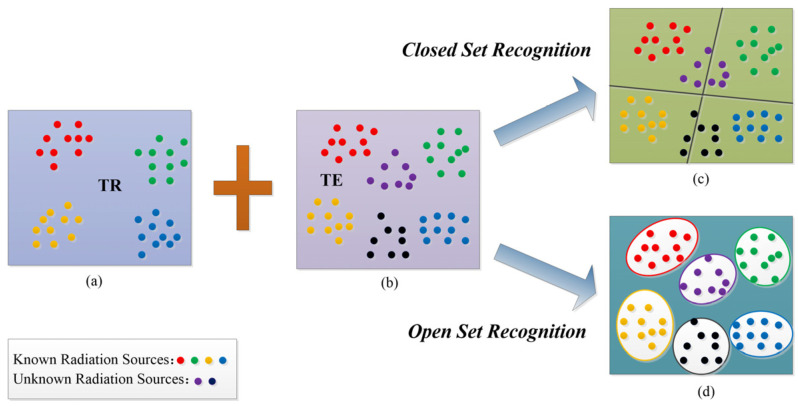
Closed set recognition and open set recognition of space radiation source. (**a**): Training Set TR. (**b**): Testing Set TE. (**c**): Closed Set Recognition. (**d**): Open Set Recognition.

**Figure 2 sensors-23-04708-f002:**
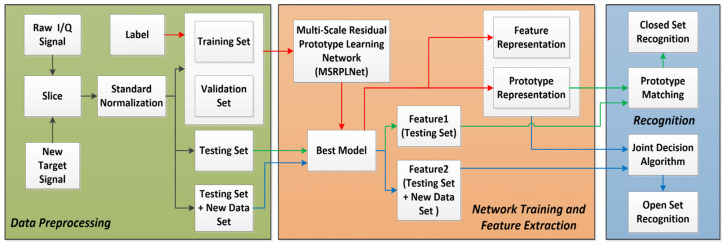
The framework of the proposed method.

**Figure 3 sensors-23-04708-f003:**
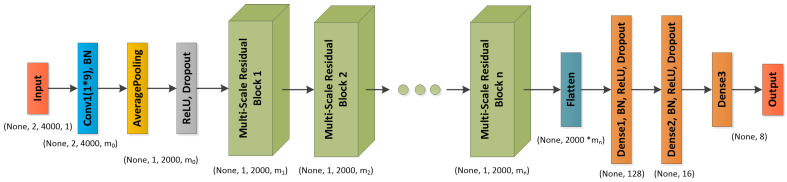
Proposed MSRPLNet.

**Figure 4 sensors-23-04708-f004:**
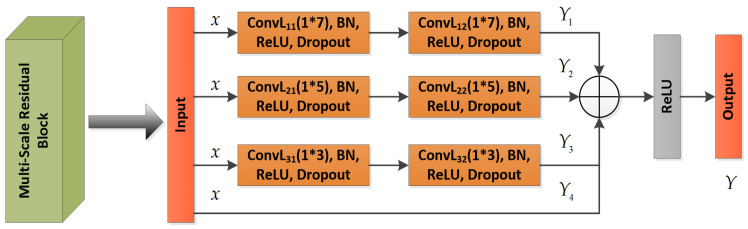
Structure of multi-scale residual block.

**Figure 5 sensors-23-04708-f005:**
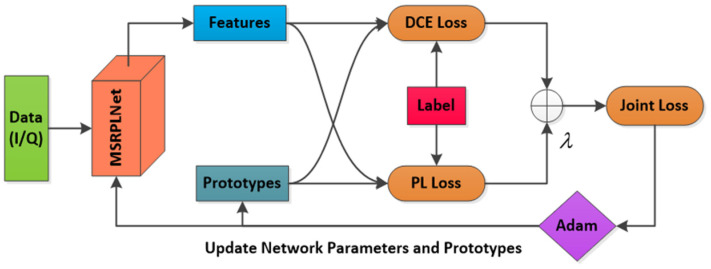
Prototype Learning Strategy.

**Figure 6 sensors-23-04708-f006:**
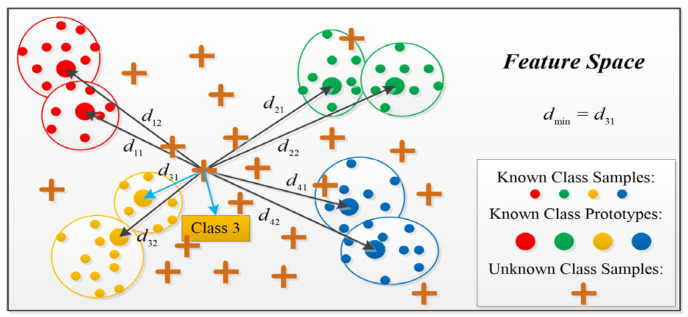
Prototype matching method.

**Figure 7 sensors-23-04708-f007:**
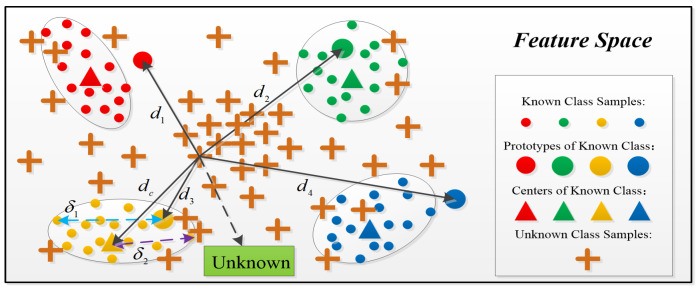
Joint decision algorithm.

**Figure 8 sensors-23-04708-f008:**
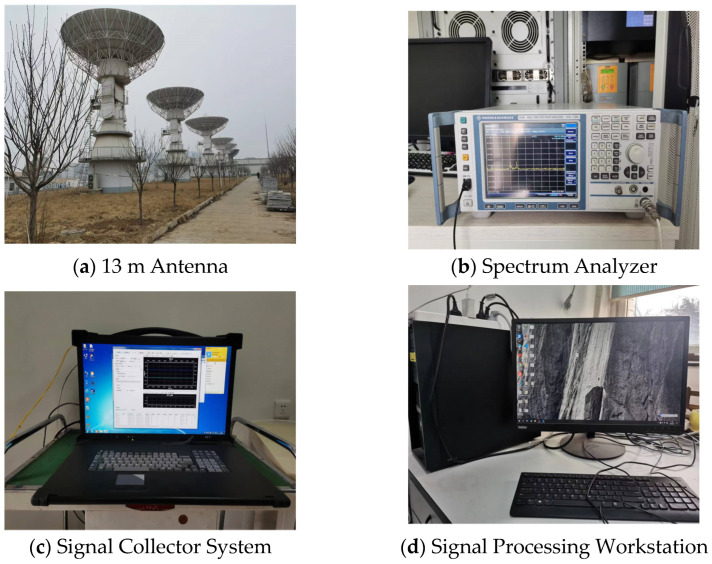
The main equipment of the satellite signal observation and receiving system.

**Figure 9 sensors-23-04708-f009:**
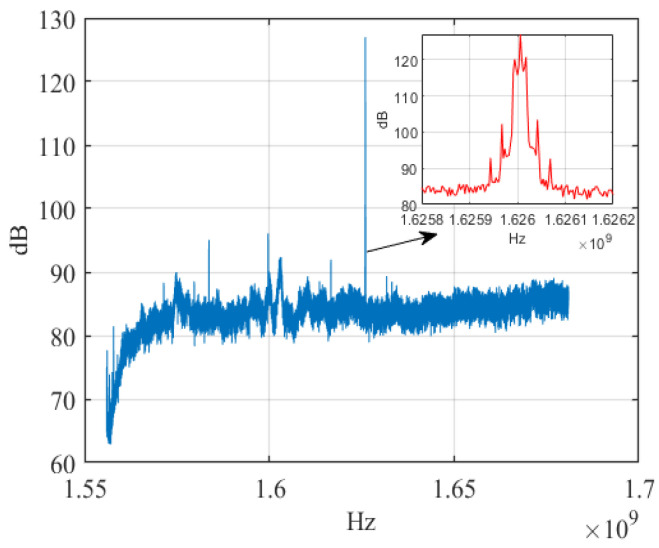
The spectrogram of Iridium STL signal.

**Figure 10 sensors-23-04708-f010:**
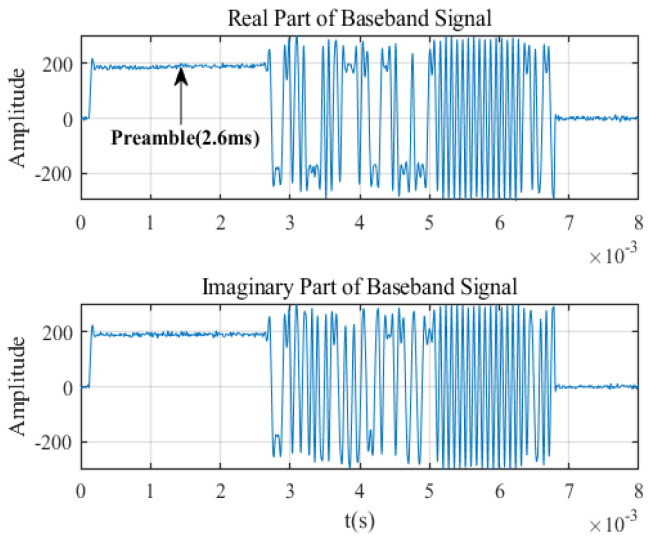
The baseband signal of Iridium STL signal.

**Figure 11 sensors-23-04708-f011:**
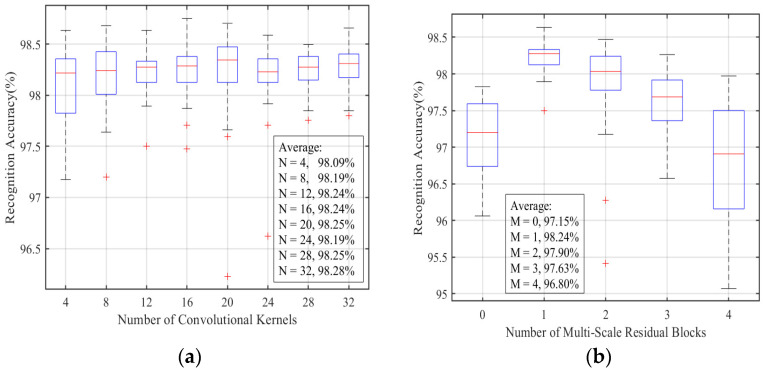
Performance comparison of different *N* and *M.* (**a**): Recognition accuracy of different convolutional kernels. (**b**): Recognition accuracy of different multi-scale residual blocks.

**Figure 12 sensors-23-04708-f012:**
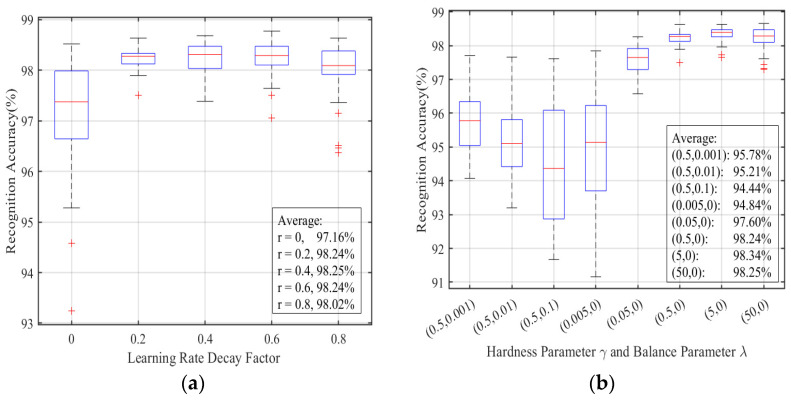
Performance comparison of different r and (γ,λ). (**a**): Recognition accuracy of different learning rate decay factor. (**b**): Recognition accuracy of different hardness parameter and balance parameter.

**Figure 13 sensors-23-04708-f013:**
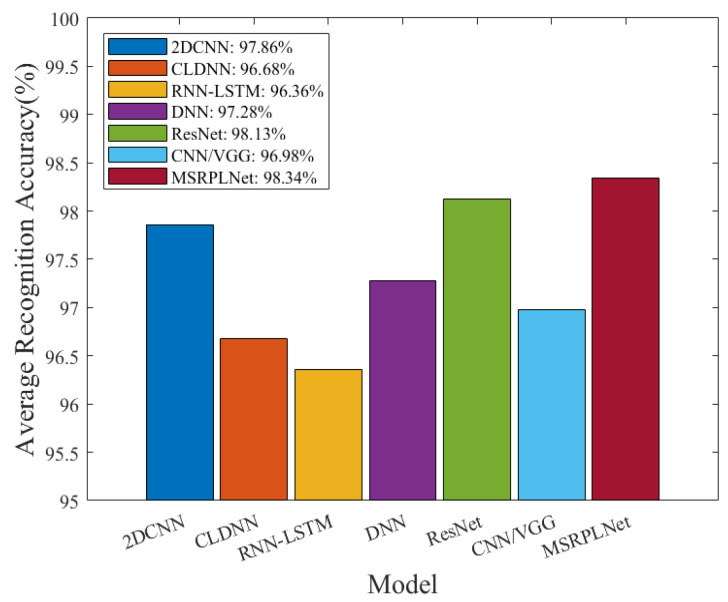
Performance comparison of 2DCNN [14], CLDNN [18], RNN-LSTM [22], DNN [28], ResNet [37], CNN/VGG [38] and MSRPLNet [Ours].

**Figure 14 sensors-23-04708-f014:**
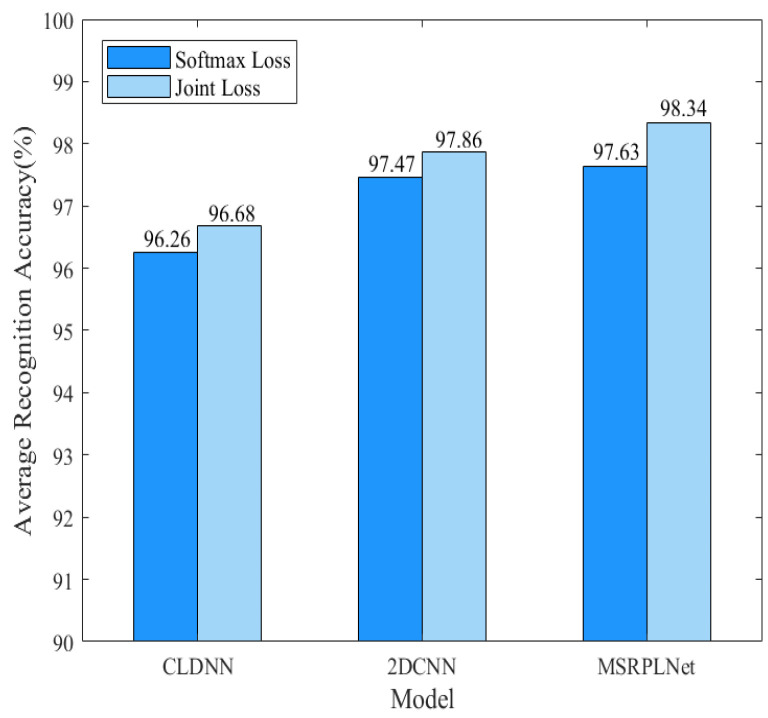
Performance comparison of different Losses.

**Figure 15 sensors-23-04708-f015:**
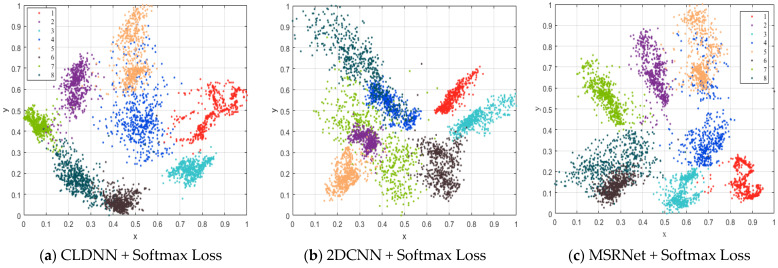
Two-dimensional scatter map of features and prototypes by different models and losses (PCA).

**Figure 16 sensors-23-04708-f016:**
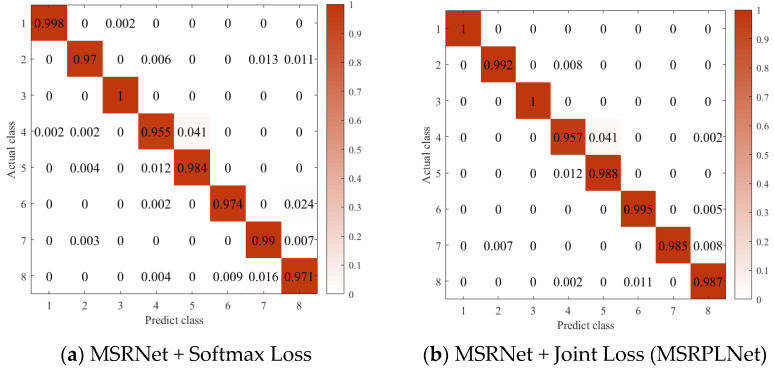
Confusion matrix of different losses.

**Figure 17 sensors-23-04708-f017:**
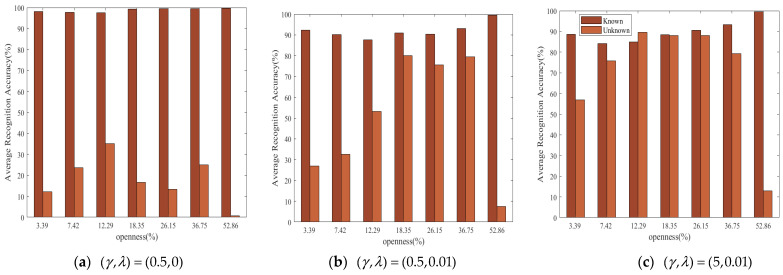
Recognition accuracy of known and unknown Iridium targets under different *openness*.

**Figure 18 sensors-23-04708-f018:**
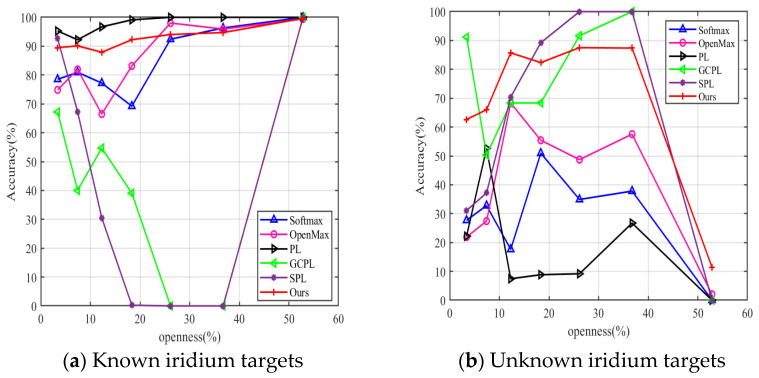
Recognition accuracy of known and unknown Iridium targets under Softmax [23], PL [24], GCPL [24], SPL [24], OpenMax [26] and Joint Decision Method [Ours].

**Figure 19 sensors-23-04708-f019:**
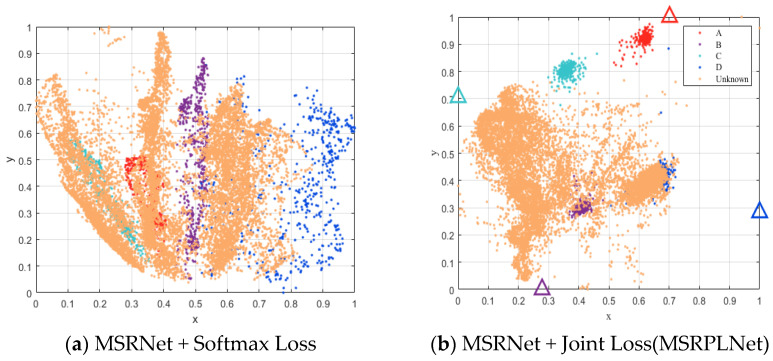
Two-dimensional scatter map of known and unknown features and prototypes (PCA).

**Figure 20 sensors-23-04708-f020:**
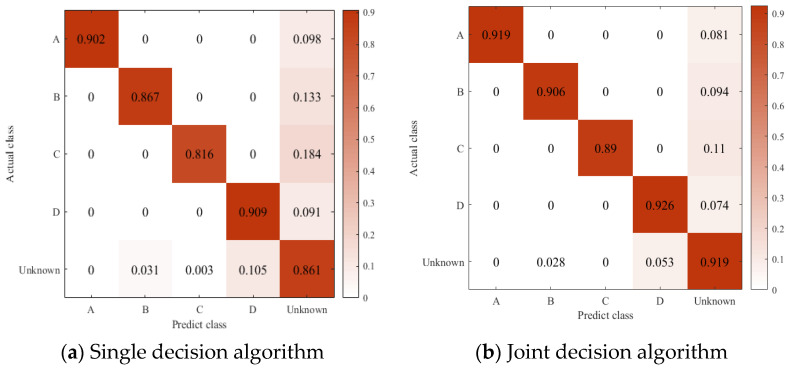
Confusion Matrix of different decision algorithms.

**Table 1 sensors-23-04708-t001:** Iridium system and Iridium STL signal.

**Orbital Altitude**	780 km (Low Earth Orbit Satellite)
**Signal Type**	Burst Signal
**Carrier Frequency**	1626~1626.5 MHz (L band)
**Symbol Rate**	25 K symbol/s
**Modulation Format**	Tone (No Modulation) + Unique Word(BPSK) + Useful Information(QPSK)
**Signal Duration**	6.5~20.32 ms
**Signal Function**	Navigation, Positioning and Timing

**Table 2 sensors-23-04708-t002:** Training parameters of MSRPLNet.

**Loss Function**	Joint Loss
**Convolutional Kernels** N	12
**Multi-Scale Residual Blocks** M	1
**Leaning Rate Decay Factor** r	0.2
**Hardness Parameter** γ	5
**Balance Parameter** λ	0
**Prototypes**	1
**“Early Stopping”**	10

**Table 3 sensors-23-04708-t003:** Time consumption of different models.

Model	Time(s)
DNN [28]	5.06
2DCNN [14]	12.54
RNN-LSTM [22]	8.71
ResNet [37]	17.38
CNN/VGG [38]	18.23
CLDNN [18]	24.87
MSRPLNet[Ours]	24.74

**Table 4 sensors-23-04708-t004:** Recognition accuracy of different (γ,λ) and *openness*.

*openness*	(0.5,0)	(0.5,0.001)	(0.5,0.01)	(5,0)	(5,0.001)	(5,0.01)
**3.39%**	55.14%	68.20%	53.53%	56.26%	**76.06%**	72.75%
**7.42%**	60.73%	68.71%	76.24%	60.75%	78.12%	**79.95%**
**12.29%**	65.21%	82.08%	83.04%	66.34%	86.81%	**87.26%**
**18.35%**	57.92%	84.65%	85.48%	60.37%	87.29%	**88.20%**
**26.15%**	56.46%	80.54%	70.49%	56.63%	**90.77%**	89.33%
**36.75%**	62.31%	85.84%	61.33%	62.37%	**91.04%**	86.26%
**52.86%**	50.17%	55.01%	54.65%	50.20%	55.52%	**56.17%**

## Data Availability

Not applicable.

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
