# Peer review of "A Novel Method for Recognizing Space Radiation Sources Based on Multi-Scale Residual Prototype Learning Network"

_sensors, 2023, doi:10.3390/s23104708_

Round 1
Reviewer 1 Report
The main contribution of this paper is to propose a prototype learning framework for closed set recognition and open set recognition of space radiation sources.
Pag. 1, line 1:The paper does not present the motivation and significance of the work clearly in the introduction section.
Pag. 5, line 234: Add more information about ‘Softmax layer’.
Pag. 6, line 248: Add more information about equation (6).
Pag. 6, line 253: The authors need to explain the novelty of the presented method compared to the state of the art.
Pag. 13, fig. 8: The figure can be improved.
The english can be improved.
Reviewer 2 Report
The title should be changed to: I'd like to suggest the heading:
"A Novel Algorithm for Recognizing Space Radiation Sources Based on Multi-Scale Residual Prototype Learning"
The accuracy of this study needs to be added to the abstract in order to improve it.
Remove the first paragraph. Therefore, the authors must begin the first paragraph of the introduction with the following declaration: "Space situational awareness (SSA) refers to the ability to monitor and predict the location and behavior of objects in space, including debris and active satellites. According to Marghany 2021; Zhao et al., 2021; Zhuo et al., 2022; and Zhan et al., 2022 this information is crucial for ensuring the safety and sustainability of space operations. Therefore, space target recognition (STR) is a critical technology that enables the identification and tracking of objects in space, including potential threats such as debris or hostile satellites. Developing advanced STR capabilities is essential for maintaining the security and longevity of space missions."
Update the references using the following one cited in the above paragraph.
1- Marghany, M. (2021). Advanced Algorithms for Mineral and Hydrocarbon Exploration Using Synthetic Aperture Radar. Elsevier.
2- Zhuo, Z., Du, L., Lu, X., Chen, J., & Cao, Z. (2022). Smoothed Lv distribution based three-dimensional imaging for spinning space debris. IEEE Transactions on Geoscience and Remote Sensing, 60, 1-13. 3- Zhan, C., Dai, Z., Soltanian, M. R., & de Barros, F. P. (2022). Data‐Worth Analysis for Heterogeneous Subsurface Structure Identification With a Stochastic Deep Learning Framework. Water Resources Research, 58(11), e2022WR033241. 4-Zhao, F., Zhang, S., Du, Q., Ding, J., Luan, G., & Xie, Z. (2021). Assessment of the sustainable development of rural minority settlements based on multidimensional data and geographical detector method: A case study in Dehong, China. Socio-Economic Planning Sciences, 78, 101066. 5- Zhou, G., Deng, R., Zhou, X., Long, S., Li, W., Lin, G., & Li, X. (2021). Gaussian inflection point selection for LiDAR hidden echo signal decomposition. IEEE geoscience and remote sensing letters, 19, 1-5. In line 40 remove "In this paper, space targets refer to various satellites with different purposes." then replace by "The term "space targets" refers to a variety of satellites with various goals. These satellites can be used for communication, navigation, earth observation, and scientific research purposes. They are an essential component of modern technology and play a significant role in our daily lives Marghany 2021; and Zhan et al., 2022). Therefore, they can also be known as space radiation sources. These sources of space radiation can pose a significant risk to astronauts and spacecraft, and it is important to continue studying and monitoring them in order to ensure the safety of space missions. Additionally, understanding these sources can also provide valuable insights into the origins and evolution of our universe." Replace the line 41 to line 43 [2] by The radiated signal would then exhibit significant nonlinear distortion because the high-power amplifier incorporated into the satellite transponder commonly operates in the saturated state or a state comparable to the saturated state to ensure high work performance. This nonlinear distortion can cause interference with other satellite signals and result in poor signal quality. To mitigate this issue, engineers may use distortion or digital signal processing techniques to reduce the distortion. This work's main objective must be discussed in a separate section. This will allow readers to easily identify the purpose of the study and understand how the research was conducted to achieve that objective. Additionally, it will help to ensure that the results and conclusions are directly related to the main objective. Revise the subtitle 2.1 as Recognizing Radiation Sources Using Data-Driven and Experience-Driven Techniques There is no correlation between equations 2,3,and4. Why use the Hilbert equation in work? Here, it is necessary to quantize the source radiation. For a precise decision, equation 5 and equation 4 must be connected, and then the probability equation must be connected to the remaining equations. This is overshadowed in this essay. Further clarification from the authors on the specific methodology used to implement equations 7 to 11 would greatly enhance the reproducibility and transparency of their results. It is important for readers to have a clear understanding of the methods employed in order to assess the validity and reliability of the findings. To produce an accurate decision, Equation 18 must be based on the probability equation. Based on accurate and quantitative results, conclusions must be revised. It is important to ensure that the data used for analysis is reliable and valid to avoid erroneous conclusions. Additionally, it is crucial to consider any potential biases or limitations in the study design that may affect the accuracy of the results. At this time, the paper is not ready for publication. Prior to publication, the authors must primarily revise the paper to clarify all ambiguities and uncertainties.Author Response
Please see the attachment.

Round 2
Reviewer 1 Report
The authors responded satisfactorily to my questions.
Reviewer 2 Report
The authors have done a good revision.